# Deterministic phase slips in mesoscopic superconducting rings

I. Petković[1], A. Lollo[1], L.I. Glazman[1,2] & J.G.E. Harris[1,2]

The properties of one-dimensional superconductors are strongly influenced by topological fluctuations of the order parameter, known as phase slips, which cause the decay of persistent current in superconducting rings and the appearance of resistance in superconducting wires. Despite extensive work, quantitative studies of phase slips have been limited by uncertainty regarding the order parameter's free-energy landscape. Here we show detailed agreement between measurements of the persistent current in isolated flux-biased rings and Ginzburg–Landau theory over a wide range of temperature, magnetic field and ring size; this agreement provides a quantitative picture of the free-energy landscape. We also demonstrate that phase slips occur deterministically as the barrier separating two competing order parameter configurations vanishes. These results will enable studies of quantum and thermal phase slips in a well-characterized system and will provide access to outstanding questions regarding the nature of one-dimensional superconductivity.

[1] Department of Physics, Yale University, 217 Prospect Street, New Haven, Connecticut 06520, USA. [2] Department of Applied Physics, Yale University, 15 Prospect Street, New Haven, Connecticut 06520, USA. Correspondence and requests for materials should be addressed to I.P. (email: ivana.petkovic@yale.edu).

Phase slips are topological fluctuations of the order parameter in one-dimensional superconductors[1]. They are responsible for the emergence of finite resistance in the superconducting state and for the decay of supercurrent in a closed loop[2–4]. Despite extensive research and a good understanding of their basic features, there remain a number of open questions related to their dynamics[5]. One of the conceptually simplest systems in which to study phase slips is an isolated, flux-biased ring. Such a system can access several metastable states and undergoes a phase slip when it passes from one of these states to another[2]. Tuning the free-energy barrier between the states to zero with the applied flux $\Phi$ will result in a deterministic phase slip from the state that has become unstable[6], whereas tuning the barrier to a small but non-zero value will lead to a stochastic phase slip via thermal activation[2,3] or quantum tunnelling[7–14].

The interpretation of measurements of stochastic phase slips[7,15–25] has been complicated by these processes' strong dependence on the system's details, such as the form of the free-energy landscape, the damping of the order parameter and the noise driving its fluctuations. Of particular importance is accurate knowledge of the barrier between metastable states, which enters exponentially into the rate of stochastic phase slips[5]. In contrast, deterministic phase slips are governed solely by the form of the free-energy landscape: they occur when the barrier is tuned to zero. For a strictly one-dimensional ring (in which the order parameter only varies along the ring's circumference), Ginzburg–Landau (GL) theory can be used to analytically calculate the barrier height, the flux at which the deterministic phase slips occur[2,3] and the measurable properties of the metastable states, for example, their persistent current[2,3,26] and heat capacity[27]. As a result, measurements of these properties that demonstrate precise agreement with theory are important for benchmarking a system in which to study thermal and quantum stochastic phase slips. Previous measurements of persistent current $I(\Phi)$ in isolated superconducting rings have found quantitative agreement with theory only at low magnetic field and very close to the critical temperature $T_c$, where metastability is absent or nearly absent[26,28,29]. However, at lower temperatures, where metastability is well-established, only qualitative agreement with theory has been demonstrated[30–32].

Here we present measurements of $I(\Phi)$ in isolated superconducting rings for temperatures spanning $T_c/2 < T < T_c$. The results, over the full range of magnetic field, show quantitative agreement with the GL theory augmented by the empirical two-fluid model[4]; the latter states the temperature dependence of the input parameters of GL theory in a broad temperature domain. The combination of the GL theory, nominally valid only at $T \to T_c$, with the two-fluid model has been shown to accurately represent the results of microscopic theory down to $T \approx T_c/2$ and was successfully used, for example, in explaining measurement of the parallel critical field of thin Al films[33] in this temperature range. We find that phase slips occur at the flux values predicted by GL theory, even to the point of demonstrating a small correction due to the rings' finite circumference[34,35]. In addition, we find that the dynamics of the phase slips is strongly damped, so that the disappearance of a barrier leads the system to relax to the adjacent local minimum. The measurement described here employs cantilever torque magnetometry, which has been shown to be a minimally invasive probe of persistent current in isolated metal rings[36] and is capable of resolving individual phase slips in a single ring[37]. As a result, these measurements demonstrate the essential features for studying stochastic phase slips: samples with a well-characterized free-energy landscape and a detection scheme suitable for measuring their intrinsic dynamics.

## Results

**Description of the system.** In this experiment, four separate samples were measured. Each sample consists of an array of 100–1,000 nominally identical aluminum rings. Arrays were used to get a better signal-to-noise ratio. Ring radii of the four samples are $R = 288–780$ nm, with nominal widths of $w = 65–80$ nm and thickness $d = 90$ nm. Detailed sample properties are listed in the Methods section and in the Supplementary Table 1. Scanning electron microscopy photos of the sample are shown in Fig. 1a.

The measurement setup is shown in Fig. 1b. A uniform magnetic field of magnitude $B$ is applied normal to the rings' equilibrium orientation. As the cantilever oscillates, current circulating in the rings experiences a torque gradient, which shifts the cantilever's resonant frequency by an amount $df$, monitored by driving the cantilever in a phase-locked loop. More details on the measurement setup are given elsewhere[37,38]. In the configuration used here, $df = \kappa\, I\, \Phi$, where $\Phi = B\pi R^2$ and $\kappa$ is a constant depending on the cantilever parameters, inversely proportional to the spring constant[36,37]. A detailed description of the conversion of data from $df$ to $I$ is given in the Supplementary Notes 1 and 2, and Supplementary Fig. 1.

**Metastable states and hysteresis.** A superconducting ring is considered one dimensional if its lateral dimensions are smaller than the coherence length $\xi$ and the penetration depth $\lambda$. The equilibrium properties of such a ring have three distinct temperature regimes, which are set by $R/\xi(T)$. For temperature $T$ only slightly below $T_c$ such that $2R < \xi$, the ring is in a superconducting state for some values of $\Phi$, whereas for the other values it is in the normal state[39,40], due to competition between the superconducting condensation energy and the flux-imposed kinetic energy of the supercurrent. At slightly lower $T$ (such that $\xi < 2R < \sqrt{3}\xi$), the condensation energy is slightly larger and for each value of $\Phi$ the ring has exactly one superconducting state. Finally, at even lower $T$ such that $2R > \sqrt{3}\xi$, the condensation energy is high enough to allow for several equilibrium states at a given $\Phi$. Depending on the ring's circumference, these three regimes may occur in the vicinity of $T_c$ described by the GL theory or may extend to lower temperatures, prompting the use of the empirical two-fluid model along with GL.

Figure 1c–e shows $I(B)$ for the sample with $R = 538$ nm as $T$ is varied. The red points show measurements taken while $B$ is increasing and the blue points while $B$ is decreasing. All the measurements exhibit sawtooth-like oscillations whose period is inversely proportional to the ring area $\pi R^2$. The smooth parts of the sawtooth represent current $I_n$ in equilibrium states characterized by the order parameter winding number $n$ and the jumps correspond to phase slips between these states. The jumps occur with flux spacing equal to the superconducting flux quantum $\Phi_0 = h/2e$, indicating that $n$ changes by unity at each jump. Measured $I(B)$ curves for all other temperatures and ring sizes are given in Supplementary Fig. 2. The three qualitative regimes described previously are accessed by varying either $T$ or $B$, as they both diminish the condensation energy. For low $T$ and $B$ the data are hysteretic, indicating the presence of multiple equilibrium states. At sufficiently high $T$ or $B$ the hysteresis vanishes, indicating that only one superconducting state is available. For the highest values of $B$ and $T$ there are ranges of $B$ over which $I = 0$ (to within the resolution of the measurement), corresponding to the rings' re-entry into the normal state. In this so-called Little–Parks regime we observe the expected features: the persistent current goes through zero when the flux bias equals an integer number of flux quanta, whereas the winding number changes at half-integer values[39,40]. This is

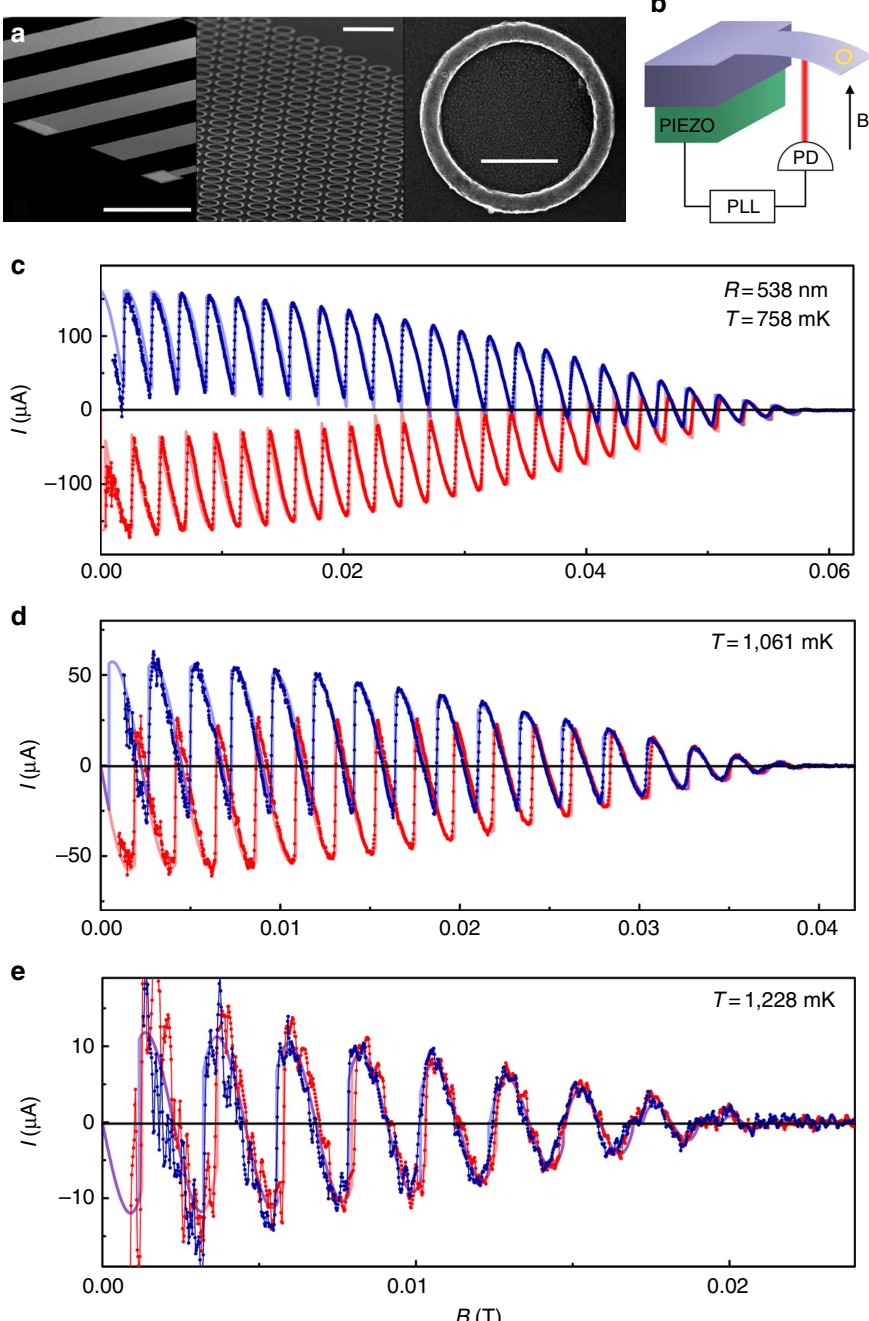

**Figure 1 | Measured I(B).** (**a**) Scanning electron microscopy photos of the sample. Left to right: several free-standing cantilevers, an array of rings on a single cantilever, a single Al ring from the array. The scale bars left to right are: 100 μm, 2 μm and 500 nm. (**b**) Measurement setup. A cantilever supporting rings is placed in a perpendicular magnetic field B. The cantilever's position is monitored by a laser interferometer (red). The signal from the photodiode (PD) is sent to a phase-locked loop (PLL), which drives a piezoelectric element (green) under the cantilever. The current in the rings is determined from the frequency of the PLL drive. (**c–e**) Supercurrent per ring I as a function of magnetic field B for rings with radius R = 538 nm at different temperatures T (marked on each panel). Points are data; thick curves are the fits described in the text. Red (blue) corresponds to increasing (decreasing) B.

described in more detail in Supplementary Note 3 and shown in Supplementary Figs 3 and 4.

**Fit to theory**. To compare these measurements with theory, we first identify the winding number n of each smooth portion of I(B). Next, we simultaneously fit all of the smooth portions of I(B) using the analytic expression derived from the GL theory for one-dimensional rings[26]. This expression includes the rings' finite

width w, which accounts for the magnetic field penetration into the ring volume and is crucial for reproducing the overall decay of I at large B. At each value of T, the fitting parameters are $\xi$ and the Pearl penetration depth $\lambda_P = \lambda^2/d$, appropriate when the bulk penetration depth $\lambda > d$ (ref. 4; ref. 41) which holds. The cantilever spring constant is assumed to be temperature independent and is used as a global fit parameter for each sample, along with the ring dimensions w and R. The resulting fits are shown as thick curves in Fig. 1c–e. The full set of fits to

measured $I(B)$ for all $R$ and $T$ is shown in Supplementary Figs 5 and 6, along with a more detailed description of the fitting procedure given in Supplementary Note 4.

In each data set we identify the rings' critical field $B_{c3}$, which we take to be the value of $B$ at which $I$ becomes indistinguishable from 0 and remains so for all $B > B_{c3}$. It is noteworthy that the identification of $B_{c3}$ is independent of any theoretical model. Next, we use the GL result for one-dimensional rings[31] $B_{c3} = 3.67\Phi_0/(2\pi w \xi(T))$ to extract $\xi(T)$ (the fit parameters are $B_{c3,0} \equiv 3.67\Phi_0/(2\pi w \xi_0)$ for each sample and $T_c$ common to all the samples). The coherence lengths $\xi(T)$ extracted from the fits of $I(B)$ and from the $B_{c3}(T)$ data agree with each other in the entire temperature interval and are approximated remarkably well by $\xi(T) = \xi_0 \sqrt{(1+t^2)/(1-t^2)}$, where $t = T/T_c$. The same relation inspired by the two-fluid model[4] was used successfully to treat the thin-film upper critical field[33,42]. Along with $\xi(T)$, fits of $I(B)$ yield the temperature dependence of the Pearl penetration depth, which agrees well with the two-fluid model, $\lambda_P(T) = \lambda_{P0}/(1 - t^4)$. Figure 2 shows the best-fit parameters $\xi$ and $\lambda_P$, as well as $B_{c3}$, all as a function of $T$. The best-fit values of $\xi_0$ ($\sim 200$ nm), $\lambda_{P0}$ ($\sim 100$ nm), $T_c$ ($\sim 1.32$ K) and $B_{c3,0}$, along with more details, are given in Supplementary Note 5 and Supplementary Table 1. Lastly, we note that $B_{c3}(T)$ should be independent of $R$ and proportional to $1/w$, consistent with the data in Fig. 2c.

**Criterion for deterministic phase slip**. Figure 1c–e shows that on each branch $I_n$, the values of current at which the phase slips occur for increasing and decreasing $B$ are located nearly symmetrically around zero current. To examine the locations of

these phase slips quantitatively, we define $\Delta\phi_n^{\pm} = \phi_n^{\pm} - \phi_{\min,n}$. Here $\phi_n^{\pm}$ is the experimental value of the normalized flux $\phi = \Phi/\Phi_0$ at which the transition $n \rightleftarrows n \pm 1$ occurs and $\phi_{\min,n}$ is the value of $\phi$ at which $I_n$ reaches zero. Flux $\phi_{\min,n}$ is either directly measured or obtained by extrapolation between sweep-up and sweep-down branches. As defined, $\Delta\phi_n^{+}$ are positive (increasing $B$, for which $n \rightarrow n+1$) and $\Delta\phi_n^{-}$ are negative (decreasing $B$, for which $n \rightarrow n-1$). (In the following we normalize all flux values by $\Phi_0$ and denote them by the character $\phi$.)

Our next step is to compare the experimental values of switching flux $\Delta\phi_n^{\pm}$ with theory. In the Langer–Ambegaokar picture, valid for a current-biased wire much longer than $\xi$, the barrier between states $n$ and $n-1$ vanishes when the bias current reaches the critical current $I_c$ (ref. 2). In the case of a flux-biased ring, still for $R \gg \xi$, the barrier between states $n$ and $n \pm 1$ goes to zero at flux values

$$\phi_{c,n}^{\pm} = \phi_{\min,n} \pm \frac{R}{\sqrt{3}\xi} + O\left(\left(\frac{w}{R}\right)^2\right), \qquad (1)$$

where $\phi_{\min,n} = \frac{n}{1 + \left(\frac{w}{2R}\right)^2}$. In the case $R \gtrsim \xi$, which corresponds to our experimental situation, it was shown that the system remains stable beyond $\phi_{c,n}^{\pm}$ and loses stability at a flux[34,35]

$$\phi_{f,n}^{\pm} = \phi_{\min,n} \pm \frac{R}{\sqrt{3}\xi} \sqrt{1 + \frac{\xi^2}{2R^2}} + O\left(\left(\frac{w}{R}\right)^2\right). \qquad (2)$$

From these expressions we see that the switching flux is set by the ratio $R/\xi$ and therefore the precise determination of $\xi$ is crucial for quantitative comparison with theory. To simplify this

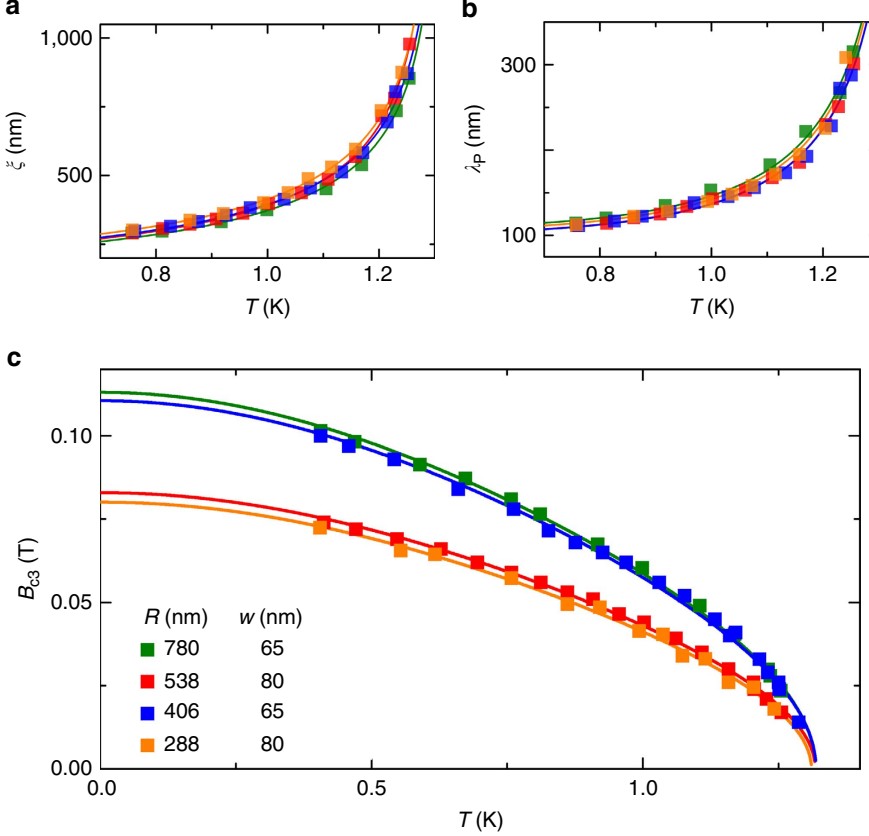

**Figure 2 | Coherence length, penetration depth and rings' critical field. (a,b)** Coherence length $\xi$ and Pearl penetration depth $\lambda_P$ as a function of temperature. The squares are the best-fit values from the GL fits described in the text. **(c)** Rings' critical field $B_{c3}$ as function of temperature. The squares are determined from measurements of $I(B)$. The lines in all panels are the fits described in the text.

comparison, it is convenient to refer all quantities not to zero field, but to the zero current field of each winding number, so we define $\Delta\phi_{c,n}^{\pm} = \phi_{c,n}^{\pm} - \phi_{min,n}$ and $\Delta\phi_{f,n}^{\pm} = \phi_{f,n}^{\pm} - \phi_{min,n}$. Additional details on the free energy landscape close to the phase slip points are given in Supplementary Note 6 and Supplementary Fig. 7.

Figure 3 shows the measured $\Delta\phi_n^{\pm}$ as a function of $n$. The vertical axis in Fig. 3 is normalized to $\Delta\phi_{f,0}^{+}$. The horizontal axis is normalized to the experimentally observed maximum winding number $n_{max}$, where $n_{max} \approx \frac{\sqrt{3}R^2}{w\xi}$. The ratio $n/n_{max}$ is very close to $B/B_{c3}$. There is a symmetry $\Delta\phi_n^{+} = -\Delta\phi_{-n}^{-}$ for $-n_{max} \leq n \leq n_{max}$; thus, it suffices to consider $n \geq 0$. Figure 3a shows the data for $R = 288$ nm. The bars represent the width of the steep portion of the sawtooth oscillations, primarily due to the small size inhomogeneities in the array (see Supplementary Figs 8 and 9, and Supplementary Notes 7 and 8). In Fig. 3b we show the data for all four samples, normalized such that all the data collapse together. Supplementary Fig. 8 shows the same data separated into four panels by ring size for a more detailed comparison.

The solid lines in Fig. 3 show the predicted $\Delta\phi_{f,n}^{\pm}/\Delta\phi_{f,0}^{+}$ (see equation (2)), whereas dotted lines in Fig. 3a show $\Delta\phi_{c,n}^{\pm}/\Delta\phi_{f,0}^{+}$ (equation (1)). The difference between the solid and dotted lines increases with the ratio $\xi(T)/R$ and is therefore the most pronounced for small rings (Fig. 3a) or at high temperature due to the increase of $\xi(T)$. We see that the prediction $\Delta\phi_{f,n}^{\pm}/\Delta\phi_{f,0}^{+}$, which includes the finite-circumference effect ($R \gtrsim \xi$), agrees well with the measured switching locations over the full range of $T$, $B$ and $R$.

The finite-circumference effect can also be seen directly in Fig. 4, which shows $I(B)$ over a narrow range of $B$ for the smallest rings. For both increasing $B$ (red) and decreasing $B$ (blue), each sawtooth oscillation reaches a maximum current and then starts to diminish before the switching occurs, as seen in the regions indicated by the black arrows.

**Damping.** For $T$ well below $T_c$ once $\xi$ is exceeded sufficiently by the circumference of the ring, there are typically multiple free-energy minima into which the system may relax. Despite this freedom, we find that the winding number always changes as $|\Delta n| = 1$. This is seen for all measured rings and all $T$ down to the lowest value $T = 460$ mK. In contrast, previous experiments[30,31] with Al rings at $T < 400$ mK have found $|\Delta n| > 1$.

We expect the tendency for $|\Delta n| > 1$ to increase with lowering $T$. Indeed, a circulating current of almost-critical value and temperature $T$ close to $T_c$ result, respectively, in the suppression of the BCS (Bardeen-Cooper-Schrieffer) singularity in the electron

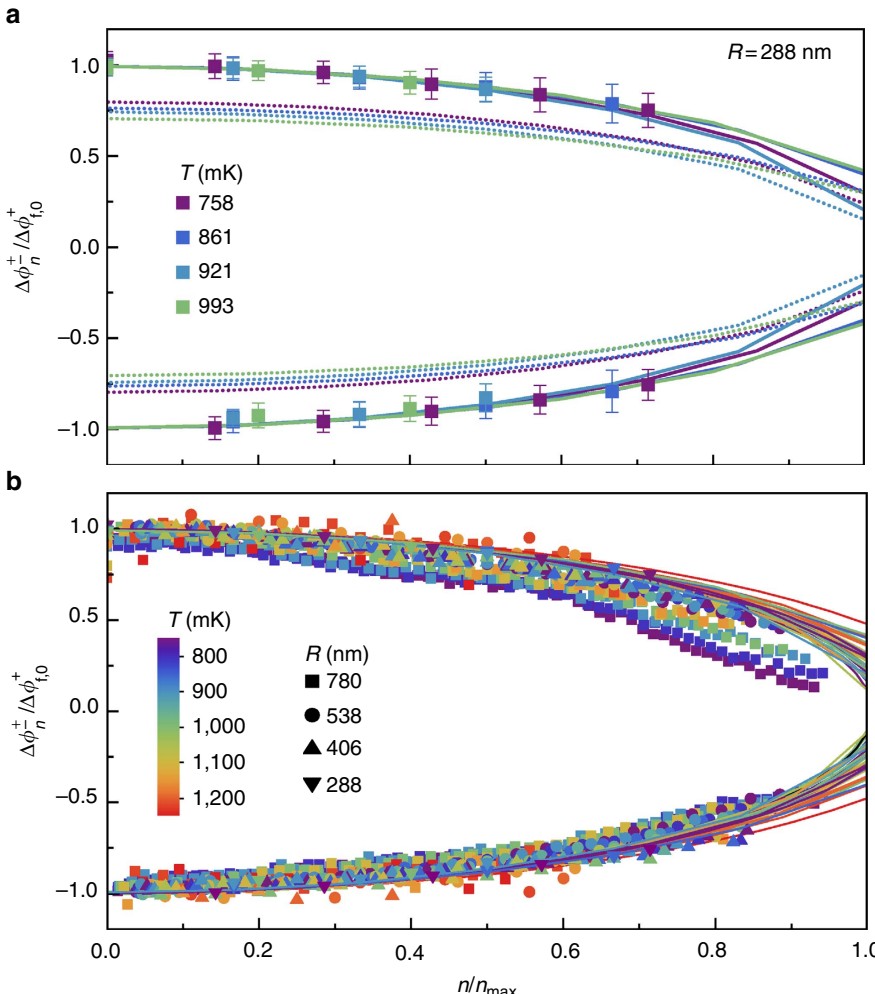

**Figure 3 | Phase slip flux as a function of winding number.** Dots: experimental values; bars in **a**: observed width of each jump due to size inhomogeneities in the array; full lines: prediction for the phase slip flux $\Delta\phi_{f,n}^{\pm}$; dotted lines in **a**: prediction for the phase slip flux $\Delta\phi_{c,n}^{\pm}$ (see text). Colours represent temperature. (**a**) The sample with $R = 288$ nm and (**b**) data from all the samples. The normalization of the axes is explained in the text.

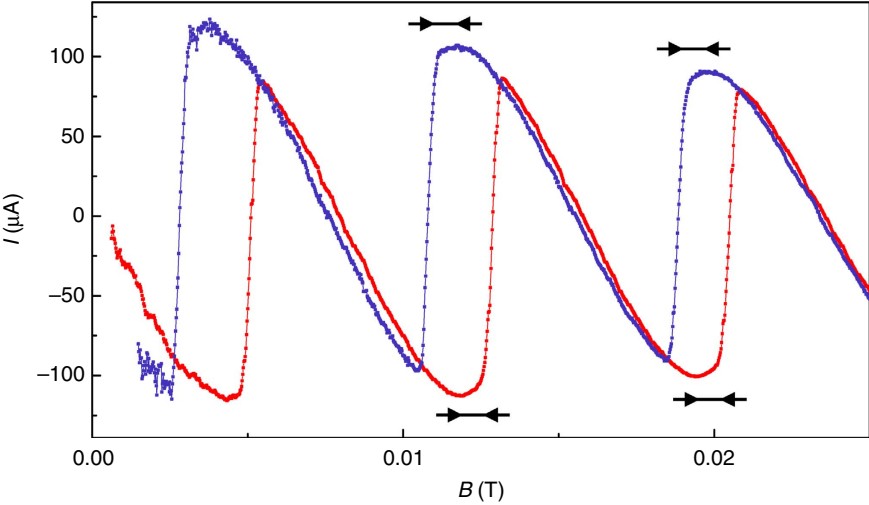

**Figure 4 | Direct observation of the finite-length correction to the phase slip criterion.** Supercurrent per ring $I$ as function of magnetic field $B$ for rings with radius $R = 288$ nm and temperature $T = 861$ mK. Red (blue) points: increasing (decreasing) $B$. The regions over which $I(B)$ diminishes at fixed winding number are indicated by black arrows. Diminishing of current after having reached a maximum but before the phase slip event is due to the finite-length correction to the phase slip criterion.

density of states and high density of Bogoliubov quasiparticles in a superconductor[4]. These are the two conditions making the dynamics of the order parameter dissipative and well described[43] by the time-dependent GL equation. In the context of phase slips[3], it determines a viscous motion of the phase difference across the phase slip, $\varphi(\tau)$ ($\tau$ being time), down the monotonic part of the effective potential relief $V(\varphi)$, and this viscous motion results in $|\Delta n| = 1$. In the opposite limit of low temperatures, the quasiparticle density is low and we may try considering the phase slip dynamics in terms of the Andreev levels associated with the phase slip. Their time evolution caused by the variation of $\varphi(\tau)$ results in Landau–Zener tunnelling between the occupied and empty levels, thus leading to dissipation[44] of the kinetic energy of the condensate (the energy is irreversibly spent on the production of quasiparticles). Our estimate (Supplementary Note 9) of the energy lost in this way is $E_{\text{diss}} \sim (\hbar S/e^2 \rho \xi)\Delta$, where $\rho$ and $S$ are, respectively, the normal state resistivity and cross-section of the aluminium wire forming the ring, and $\Delta$ is the superconducting gap; a numerical proportionality factor is beyond the accuracy of the estimate.

The condensate energy difference between the two metastable states involved in a $|\Delta n| = 1$ transition is $E_{\Delta n = 1} = (\hbar/e)j_c S \sim (\hbar S/e^2 \rho \xi)\Delta$; here, $j_c \sim \Delta/(e\rho\xi)$ is the critical current density. Furthermore, the lower of the two states is protected by a barrier $\delta F_{\Delta n = 1} \sim (\xi/R)^{5/2} E_{\Delta n = 1}$ (the estimate is easily obtained from the Langer–Ambegaokar[2] scaling, $\delta F \propto (1 - j/j_c)^{5/4}$, of the barrier with the current density $j$, see Supplementary Note 9). The height of the barrier is smaller for larger rings.

We find the irreversibly lost energy $E_{\text{diss}}$ to be of the order of the energy difference between the two metastable states $E_{\Delta n = 1}$. The above estimates, given their limited accuracy, allow (but do not guarantee) the condensate to have a sufficient excess of kinetic energy to overcome a small barrier out of the metastable state with $\Delta n = 1$. In addition to higher temperatures, in a notable difference from the previous experiments the rings studied here had smaller $R$, providing a better protection of the metastable states.

## Discussion

We have studied the persistent current in arrays of flux-biased uniform one-dimensional superconducting Al rings. We found detailed agreement with GL theory, including the location of deterministic phase slips, which are predicted to occur when the

barrier confining the metastable state occupied by the ring goes to zero. In one dimension, GL theory has a relatively simple, analytic form and, due to their small width, our rings are strictly in the one-dimensional limit, in contrast to those studied previously[30–32]. As a result, GL theory provides detailed knowledge of the free-energy landscape in these samples. This should enable systematic study of thermal and quantum phase slips in isolated rings and progress towards the quantitative understanding of coherent quantum phase slips[45,46], one of the outstanding goals in the field[14,47,48].

## Methods

**Sample fabrication.** Ring radii of the four measured samples are $R = 288$, 406, 538 and 780 nm, nominal widths are $w = 65$ nm (for $R = 406$, 780 nm) and 80 nm (for $R = 288$, 538 nm), and thickness $d = 90$ nm. Further details on sample properties are listed in the Supplementary Table 1. Each array is fabricated on a Si cantilever of length $\sim 400\,\mu$m, thickness 100 nm and width $\sim 60\,\mu$m, with resonant frequency $f \sim 2$ kHz, spring constant $k \sim 1$ mN m$^{-1}$ and quality factor $Q \sim 10^5$. Cantilevers are fabricated out of a silicon-on-insulator wafer. They are patterned out of the top silicon layer by means of optical lithography followed by a reactive ion etch. Rings are then fabricated on top of patterned cantilevers using standard e-beam lithography with a polymethyl methacrylate (PMMA) mask, into which Al is evaporated in a high-vacuum thermal evaporator. After lift-off, the top of the wafer is protected and the backing silicon layer is etched in KOH, followed by a BOE etch of the SiO$_2$ layer and drying in a critical point dryer. This results in cantilevers being fully suspended. Further details on the the fabrication process are given elsewhere[36,37].

**Data availability.** The data that support the findings of this study are available from the corresponding author upon request.

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

## Acknowledgements

We thank Amnon Aharony, Richard Brierley, Michel Devoret, Ora Entin-Wohlman, Alex Kamenev, Konrad Lehnert, Hendrik Meier and Zoran Radović for useful discussions, and Ania Jayich and Will Shanks for fabricating the samples. We acknowledge support from the National Science Foundation (NSF) Grant Number 1106110 and the US-Israel Binational Science Foundation (BSF). L.G. was supported by DOE contract DEFG02-08ER46482.

## Author contributions

A.L. and I.P. performed the measurement. All authors conducted the analysis. I.P., J.G.E.H. and L.I.G. wrote the manuscript. All authors discussed the results and commented on the manuscript.

## Additional information

**Competing financial interests:** The authors declare no competing financial interests.

**Publisher's note**: 

