## [Peer Review File · Nature Communications]

Reviewers' comments:

Reviewer #1 (Remarks to the Author):

The manuscript "Deterministic phase slips in mesoscopic superconducting rings" by I. Petkovic et al. describes the current oscillations in the superconducting nanorings with the change of magnetic field at different temperatures and different radius of the ring. The observed oscillations are due to topological fluctuations of the order parameter, known as phase slips. The authors identify the winding number n for each smooth part of $I(B)$ and then fit $I(B)$ to the Ginzburg-Landau (GL) theory. The fitting parameters are superconducting coherence length and Pearl penetration depth (see Fig.2 of the manuscript). These two parameters characterize the free energy landscape in the sample. The next step in the paper is the location of the flux at which the phase slip occurs and comparison it with the prediction of GL theory. According to Fig. 3 there is good agreement between the theory and experiment. According to the authors the main goal of the paper is that the results of this paper will enable studies of quantum and thermal phase slips in a well-characterized system and will provide access to outstanding questions regarding the nature of one-dimensional superconductivity. The paper is well written and definitely should be published in the specialized Journal like Physical Review B. The reason for that is simple. This paper provides preliminary results which help to study outstanding issues of quantum and thermal phase slip processes. Basically the results provide the sample characterization procedure. Secondly the coherence length as well as the London penetration depth may be studied by different technique. Therefore proposed technique is not unique. Therefore I suggest to submit the paper to the specialized Journal where it will attract the attention of the specialists working in the field.

Reviewer #2 (Remarks to the Author):

The manuscripts presents interesting and important results on deterministic phase slips. The measurements are in good agreement with GL model. The results are expected to provide a strong drive for the understanding of quantum phase slips in superconducting loops, since now the GL energy can be used in quantum phase slips calculations. I recommend the paper should be published after the following remarks are addressed:

1. " For the highest values of B 124 and T there are ranges of B over which $I = 0$ (to within 125 the resolution of the measurement), corresponding to the 126 rings' re-entry into the normal state." This is an interesting statement. Yet I could not find the corresponding regions in the provided graphs. Also, according to Little-Parks effect, zero current happens not only when the sample is normal but also when the flux equals flux quantum multiplied by an integer number. These issues need to be clarified.
2. Individual phase slips in superconducting loops have been previously observed by other groups. Those previous results should be cited. Such observations have been reported in, e.g., ["Little-Parks oscillations at low temperatures: Gigahertz resonator method", A. Belkin, M. Brenner, T. Aref, J. Ku, A. Bezryadin, Appl. Phys. Lett. 98, 242504 (2011)] and ["Quantum phase slip phenomenon in ultra-narrow superconducting nanorings", Konstantin Yu. Arutyunov, Terhi T. Hongisto, Janne S. Lehtinen, Leena I. Leino & Alexander L. Vasiliev, Scientific Reports 2, 293(1-7) (2012)].
3. It appears that λ_p is defined through λ . But there is no definition for λ itself.
4. What was the method used to make the loops? Liftoff? Reactive ion etching?
5. The symbol " d " is usually used to denote the derivative. Yet, in this manuscript it is used to denote finite differences. Perhaps it would make the paper easier to read if Δ is used as a notation for the finite difference.
6. In the Little-Parks oscillation theory, the winding number changes when the flux equals half-flux-

quantum. I wonder if in the present experiments such simple result is observed or not. If not, by how much the flux at which the phase slip occurs differs from the half-flux-quantum condition.

Reviewer #3 (Remarks to the Author):

This manuscript describes an interesting experiment. The magnetic moment of arrays of small superconducting rings has been measured by means of cantilever torque magnetometry. The rings have small dimensions and except very close to T_c they are in the one-dimensional regime (circumference longer than the coherence length, cross-section smaller). The phase winds along the length of the wire leading to the persistent current that is measured, and phase-slips lead to 2π jumps. These experiments follow in a series of previous experiments on rings by different authors with different techniques. New are the small rings and the possibility to extend to temperatures as low as $T_c/2$. The authors find quite detailed agreement with theoretical predictions. The experiments are clever, well-performed and well described. They definitely deserve to be published.

The paper is clearly written and contains extensive references to previous work.

I asked myself the question what this manuscript offers beyond a further, more detailed set of measurements that can be understood with the theory available. In the Discussion at the end of the paper the authors say that their new method 'should enable systematic study of thermal and quantum phase slips in isolated rings, and progress towards the quantitative understanding of coherent quantum phase slips, one of the outstanding goals in this field'. Can their new technique indeed be refined so that a single ring can be studied? This would certainly be a major step, but to me it is not obviously possible.

The spirit of this manuscript is very much to look for agreement with theory, but there is a certain sloppiness in the discussion. The authors in their introduction say they make a fit to Ginzburg-Landau theory. In the second paragraph below figure 2 they describe how in fact they fit to empirical expressions for coherence length and Pearl penetration depth that are based on London and two-fluid expressions and not supported by Ginzburg-Landau. It can certainly not be said that these are 'valid' for Al down to $T_c/2$, they only seem to work. I would have preferred the authors to be more open-minded, looking for new effects that are not so much predicted by theories that are shaky once the temperature goes down. An example is the question of damping, to which a separate small section is devoted. Damping is found to be more pronounced in these new small rings. No attempt at all is made to think about that. What is the mechanism of damping? Is it internal in the rings (quasiparticles, those should disappear at low temperatures; is there an effect when measurements are performed slower?) or external (what is the electromagnetic environment of an individual ring?). On the way to the understanding of coherent quantum phase slips this could be a major clue.

REVIEWERS' COMMENTS:

Reviewer #2 (Remarks to the Author):

The manuscript in its present form is ready to be published. All my remarks have been addressed in a satisfactory manner.

Reviewer #3 (Remarks to the Author):

The new version of the paper is a definite improvement. I think the authors have reacted well to the comments from reviewers in the first round.

I recommend that the paper in the present form be published.

We thank the reviewers for their comments, suggestions and questions.
We would like to proceed to reply point-by-point.

The Reviewer #1 writes:

The manuscript "Deterministic phase slips in mesoscopic superconducting rings" by I. Petkovic et al. describes the current oscillations in the superconducting nanorings with the change of magnetic field at different temperatures and different radius of the ring. The observed oscillations are due to topological fluctuations of the order parameter, known as phase slips. The authors identify the winding number n for each smooth part of $I(B)$ and then fit $I(B)$ to the Ginzburg-Landau (GL) theory. The fitting parameters are superconducting coherence length and Pearl penetration depth (see Fig.2 of the manuscript). These two parameters characterize the free energy landscape in the sample. The next step in the paper is the location of the flux at which the phase slip occurs and comparison it with the prediction of GL theory. According to Fig. 3 there is good agreement between the theory and experiment. According to the authors the main goal of the paper is that the results of this paper will enable studies of quantum and thermal phase slips in a well-characterized system and will provide access to outstanding questions regarding the nature of one-dimensional superconductivity.

The paper is well written and definitely should be published in the specialized Journal like Physical Review B. The reason for that is simple. This paper provides preliminary results which help to study outstanding issues of quantum and thermal phase slip processes. Basically the results provide the sample characterization procedure. Secondly the coherence length as well as the London penetration depth may be studied by different technique. Therefore proposed technique is not unique. Therefore I suggest to submit the paper to the specialized Journal where it will attract the attention of the specialists working in the field.

OUR RESPONSE: The reviewer summarizes our results as a "sample characterization procedure". However we respectfully disagree with this summary; the measurements presented in this paper demonstrate a number of important results that will be of interest to a broad audience. In particular, this paper is the first quantitative study of supercurrent in isolated one dimensional superconducting rings that provides a detailed comparison between data and theory in the full range of magnetic field, over a large range of temperature and ring size, and in both the reversible and irreversible regimes. It is also the first quantitative study of the deterministic phase slip criterion.

The reviewer is correct to state that other techniques for measuring the coherence length (ξ) and penetration depth (λ) of a superconductor have been available for a while, as have measurements of the supercurrent in isolated rings. However they have not been combined in a single study to provide the type of quantitative analysis shown in the present paper. The approach used in our experiment enables us to determine ξ and λ in situ, on the very sample on which the statistics of phase slips was later measured (as described in the reply to reviewer #3 below, these measurements of phase slip statistics are being written up for publication). We obtain very accurate estimates of ξ and λ , which in turn enables us to calculate the barrier confining the metastable state. In the present work this has lead to a quantitative study of the criterion for the deterministic phase slip, at which the barrier confining a metastable state goes to zero. Since for stochastic phase slips the escape rate depends exponentially on the barrier height, this approach is expected to result in a better quantitative understanding of thermal and quantum phase slips.

We also feel that this work should not be seen as a preliminary result. Rather, it provides the first quantitative analysis of $I(B)$ in a wide range of parameters, demonstrates accurate measurements of key system parameters and presents the first quantitative study of the deterministic aspect of phase slips in one-dimensional superconductors (i.e. the study of the potential landscape of metastable states and barriers between them).

The Reviewer #2 writes:

The manuscript presents interesting and important results on deterministic phase slips. The measurements are in good agreement with GL model. The results are expected to provide a strong drive for the understanding of quantum phase slips in superconducting loops, since now the GL energy can be used in quantum phase slips calculations. I recommend the paper should be published after the following remarks are addressed:

1. " For the highest values of B and T there are ranges of B over which $I = 0$ (to within the resolution of the measurement), corresponding to the rings' re-entry into the normal state." This is an interesting statement. Yet I could not find the corresponding regions in the provided graphs. Also, according to Little-Parks effect, zero current happens not only when the sample is normal but also when the flux equals flux quantum multiplied by an integer number. These issues need to be clarified.

OUR RESPONSE: We agree with the reviewer that the experimental signatures of the Little-Parks effect were not obvious in the original figures. To address this point, we have added Supplementary Note 3 to the Supplementary Material. In that note, Supplementary Figure 3 shows three sets of measurements of $I(B)$ curves that clearly show large normal state regions (denoted by black arrows) that occur well below the rings' critical field and are in between superconducting regions. Supplementary Figure 4 shows that the flux values at which supercurrent passes through zero correspond to integer numbers of flux quanta as expected. These points are now also referenced in the main text (lines 140-145).

2. Individual phase slips in superconducting loops have been previously observed by other groups. Those previous results should be cited. Such observations have been reported in, e.g., ["Little-Parks oscillations at low temperatures: Gigahertz resonator method", A. Belkin, M. Brenner, T. Aref, J. Ku, A. Bezryadin, *Appl. Phys. Lett.* 98, 242504 (2011)] and ["Quantum phase slip phenomenon in ultra-narrow superconducting nanorings", Konstantin Yu. Arutyunov, Terhi T. Hongisto, Janne S. Lehtinen, Leena I. Leino & Alexander L. Vasiliev, *Scientific Reports* 2, 293(1-7) (2012)].

OUR RESPONSE: We thank the reviewer for pointing out these references, and have added them to our paper.

3. It appears that λ_p is defined through λ . But there is no definition for λ itself.

OUR RESPONSE: In the main text we define $\lambda_p = \lambda^2/d$ and have added a statement that λ is the bulk penetration depth (line 155 in the new Main text).

4. What was the method used to make the loops? Liftoff? Reactive ion etching?

OUR RESPONSE: The method used to make loops is indeed liftoff, using standard e-beam lithography with a PMMA mask. We have re-written Methods (adding lines 338-347) to outline the sample fabrication in sufficient detail.

5. The symbol "d" is usually used to denote the derivative. Yet, in this manuscript it is used to denote finite differences. Perhaps it would make the paper easier to read if Δ is used as a notation for the finite difference.

OUR RESPONSE: We agree with the reviewer about this confusing notation, and have replaced symbol "d" with " Δ " throughout the revised paper.

6. In the Little-Parks oscillation theory, the winding number changes when the flux equals half-flux-quantum. I wonder if in the present experiments such simple result is observed or not. If not, by how much the flux at which the phase slip occurs differs from the half-flux quantum condition.

OUR RESPONSE: Changing of winding number at half-flux value in the Little-Parks region is indeed observed. This is addressed together with point 1 in lines 140-145 of the new main text and in the new Supplementary Note 3. The half-integer values of flux at which the winding number changes by one are illustrated in Supplementary Figure 4.

The Reviewer #3 writes:

This manuscript describes an interesting experiment. The magnetic moment of arrays of small superconducting rings has been measured by means of cantilever torque magnetometry. The rings have small dimensions and except very close to T_c they are in the one-dimensional regime (circumference longer than the coherence length, cross-section smaller). The phase winds along the length of the wire leading to the persistent current that is measured, and phase-slips lead to 2π jumps. These experiments follow in a series of previous experiments on rings by different authors with different techniques. New are the small rings and the possibility to extend to temperatures as low as $T_c/2$. The authors find quite detailed agreement with theoretical predictions. The experiments are clever, well-performed and well described. They definitely deserve to be published.

The paper is clearly written and contains extensive references to previous work.

I asked myself the question what this manuscript offers beyond a further, more detailed set of measurements that can be understood with the theory available.

In the Discussion at the end of the paper the authors say that their new method 'should enable systematic study of thermal and quantum phase slips in isolated rings, and progress towards the quantitative understanding of coherent quantum phase slips, one of the outstanding goals in this field'. Can their new technique indeed be refined so that a single ring can be studied? This would certainly be a major step, but to me it is not obviously possible.

OUR RESPONSE: The present setup is indeed capable of measuring individual phase slips in an individual ring. This is demonstrated in Appendix F of Reference 37 of the main text (Shanks, W. E., Persistent Currents in Normal Metal Rings, Ph.D. Thesis, Yale University, 2011). In addition, we have recently made extensive measurements of phase slips in individual rings using the same setup as in the present paper (albeit with some technical improvements in magnetic field stability). These measurements are sufficiently sensitive to produce detailed switching histograms of the phase slip events over a wide

range of temperature, magnetic field, and ring size. These results on the switching behavior in individual rings have been presented at a number of seminars, and we are in the process of writing them up for publication.

The spirit of this manuscript is very much to look for agreement with theory, but there is a certain sloppiness in the discussion. The authors in their introduction say they make a fit to Ginzburg-Landau theory. In the second paragraph below figure 2 they describe how in fact they fit to empirical expressions for coherence length and Pearl penetration depth that are based on London and two-fluid expressions and not supported by Ginzburg-Landau. It can certainly not be said that these are 'valid' for A_I down to $T_c/2$, they only seem to work.

OUR RESPONSE: We agree with the reviewer that this relationship was not clearly described in the original manuscript, and deserved further explanation. To address this point, we have made changes in the Main text (lines 48-59, 115-118, 168-184) and to the Supplementary Material (Supplementary Note 4) to outline the spirit in which we combine the two theories.

I would have preferred the authors to be more open-minded, looking for new effects that are not so much predicted by theories that are shaky once the temperature goes down. An example is the question of damping, to which a separate small section is devoted. Damping is found to be more pronounced in these new small rings. No attempt at all is made to think about that. What is the mechanism of damping? Is it internal in the rings (quasiparticles, those should disappear at low temperatures;

OUR RESPONSE: In the revised manuscript, we have added a more detailed discussion of this point in the section labelled "Damping" (in the main text, lines 261-310) and in Supplementary Note 9. To briefly summarize, we argue that at temperatures close to T_c and for a circulating current close to the critical value, the time-dependent Ginzburg Landau (TDGL) equation is applicable. Since the TDGL equation describes viscous evolution of the phase difference across the phase slip, it is expected that the phase slip dynamics will be overdamped in this regime. At lower temperatures we treat the phase slip region as an SNS junction, and compare the energy released during a phase slip to the energy dissipated in the production of quasiparticles by Landau-Zener tunneling between the Andreev levels in the corresponding junction. We find these energies to be of the same order of magnitude, and independent of the ring radius R , while the size of the barrier confining the lower-energy state decreases with increasing R . This opens the possibility of underdamped dynamics for larger rings at low temperature; this qualitative trend is consistent with the existing experimental findings.

is there an effect when measurements are performed slower?)

OUR RESPONSE: We have varied the sweep rate by two orders of magnitude and we have consistently measured the change of the winding number by one. However we are only able to administer very slow sweep rates (0.1-10G/s), well below any characteristic physical time of the process. At higher sweep rates the temperature of the fridge rises due to eddy-current heating.

or external (what is the electromagnetic environment of an individual ring?). On the way to the understanding of coherent quantum phase slips this could be a major clue.

OUR RESPONSE: The rings are electrically isolated – i.e., they are not contacted by leads and they are not coupled capacitively or inductively to any microfabricated circuit. We have measured the electron temperature in many aluminum samples mounted on similar cantilevers in the same apparatus (in both

the normal and the superconducting state), and have always found that the electron temperature tracks the refrigerator down to its lowest temperature. Of course, this is provided that standard precautions are followed (not using too much laser power in the interferometer, ensuring that the laser points at a region of the cantilever far from the rings, etc). A quantitative study of some of these issues is presented in Refs. 36 and 37 of the main text, as well as in Bleszynski-Jayich et al., *Appl. Phys. Lett.* 92, 013123 (2008). The conclusion of all these studies is that under the conditions used in the present manuscript, the electron temperature equilibrates with the fridge temperature, and that this equilibrium is reached via the phonons. Thus, we believe that the energy released by a phase slip is first dissipated to the quasiparticles (as described two paragraphs above), and that the quasiparticles then equilibrate to the fridge temperature via phonons. This picture is consistent with straightforward estimates of the thermal coupling rates between the electrons and the phonons in the aluminum, and then between phonons in the aluminum and phonons in the silicon cantilever.